# Suicidal Ideation and Severity of Distress among Refugees Residing in Asylum Accommodations in Sweden

**DOI:** 10.3390/ijerph16152751

**Published:** 2019-08-01

**Authors:** Anna Leiler, Michael Hollifield, Elisabet Wasteson, Anna Bjärtå

**Affiliations:** 1Department of Psychology and Social Work, Mid Sweden University, 831 25 Östersund, Sweden; 2War Survivors Institute, Long Beach, CA 90815, USA

**Keywords:** suicidal ideation, refugees, severity of distress

## Abstract

Refugees worldwide suffer high levels of distress and are at increased risk for death by suicide. The Refugee Health Screener (RHS) was developed to screen for emotional distress among refugees and can be used to assess distress severity. This paper examines the association between distress severity and suicidal ideation in a sample of refugees residing in asylum accommodations. Data from the RHS and item 9 on the Patient Health Questionnaire-9 (PHQ-9) was analyzed. Results showed that individuals at moderate and severe levels of distress were much more likely to exhibit suicidal ideation than individuals with low levels of distress. Even though we cannot conclude that individuals with low levels of distress do not have thoughts of ending their lives, further suicide assessment is warranted in asylum seekers with moderate to severe distress on the RHS.

## 1. Introduction

Refugees worldwide suffer from stress- and trauma-related mental health problems, such as symptoms of posttraumatic stress, anxiety and depression, many of which having symptoms at diagnostic levels [1,2,3,4]. Mental disorders are generally associated with adverse outcomes such as higher disability, use of health service, substance abuse and financial dependency [5]. Both depression [6] and posttraumatic stress disorder (PTSD) [7] are associated with increased rates of suicidal ideation and attempts. Refugees have a decreased life expectancy and worse overall health [8,9], including an increased risk for death by suicide compared to the general population [10,11]. A recent Swedish report [12] showed an increased risk of self-harm, suicide attempts and death by suicide among young refugees. Suicidal ideation, an important predictor of death by suicide [13,14,15], has been shown to be more prevalent in refugees than in a cross-national sample (27–31% vs. 9.2%) [16,17,18]. 

The high levels of distress among refugees, and the associated increased risk for suicidal ideation, calls for routine mental health screening upon arrival to a new country. The Refugee Health Screener (RHS) [19,20] was developed as a brief screener for identifying symptoms of depression, anxiety and PTSD among refugees. The RHS-13-item version [20] has recently been shown to be efficient in identifying severity of depression, anxiety and PTSD symptoms, using cutoff values to indicate mild, moderate and severe distress [21]. However, since the RHS does not include any items about suicidal ideation, investigating the ability of the RHS-13 to detect individuals with suicidal ideation is warranted. Data from a large-scale prevalence study were analyzed focusing on prevalence of suicidal ideation and risk for suicidal ideation at different levels of distress on the RHS-13. 

## 2. Materials and Methods

This study was performed as a part of the project AMIR (Assessment of Mental Health and Early Intervention for Refugees). Results from the full project have been reported previously [21,22]. An outline of the methods section is presented herein, and more detail is in previous reports. The study was approved by the regional ethical review board (Dnr: 2016-364-31).

### 2.1. Sampling Procedures and Participants

This was a cross-sectional study with a convenience sample of refugees residing in asylum accommodations in Jämtland-Härjedalen county, Sweden, during the period November 2016 to April 2017. A list of all asylees residing in the county was provided by the Swedish Migration Agency. All adult refugees (≥18 years, *N* = 1332) were invited to participate by postal letters with information about the study and a schedule for meetings where screening took place.

Members of the research team and bilingual staff, most of whom had a background in health care from their home countries (one psychologist and two physicians), visited 13 locations in the county, all housing more than 30 refugees. Participants were offered to be screened on-site using iPads, or online using a web-based software (Qualtrics, Provo, UT, 2005). Only 15 individuals (2.9%) chose to answer online. The bilingual staff helped translate when necessary, yet their role was primarily as a cultural broker or navigator [23] to help subjects with study procedures. They assisted the participants in answering the survey and provided culturally sensitive explanations of the project’s purpose. 

Five hundred and ten individuals completed the full questionnaire (367 males, 136 females, 7 identified as other), with the majority being young (363 individuals or 71.2% below 35 years of age). Most individuals came from Afghanistan (38.4%) and Syria (26.9%). Twenty-eight percent (n = 143) had already received a positive decision of their asylum application and were waiting for a decision to transfer to a municipality. 

### 2.2. Instruments

Participants completed the RHS-13 [20] and the Patient Health Questionnaire-9 (PHQ-9) [24] for the current study. The Generalized Anxiety Disorder-7 (GAD-7) [25], the Primary Care PTSD-4 (PC-PTSD-4) [26], and the World Health Organization Quality of Life – Brief (WHOQOL-BREF) [27] were completed at the same time for the larger study reported elsewhere [21,22]. The questionnaires were administered in seven different languages (Arabic, Dari, Farsi, Tigrinya, Somali, English, and Swedish). Nonliterate participants could request that the questions were read aloud to them by choosing audio support (available in the four first mentioned languages). 

The distress severity was assessed with the RHS-13 [20], each item being a five-point Likert scale (0–4; total range 0–52) with a sum of scores ≥11 indicating significant distress. A previous study [21] confirmed 11 as a cutoff for distress predictive of PTSD, anxiety, and depression, and further suggested the use of cutoffs to indicate mild (≥11), moderate (≥18), and severe (≥25) symptoms. This stratification suggests the use of immediate intervention for severe cases and further assessment and/or follow-up for moderate and mild cases.

Suicidal ideation was measured using item nine of the PHQ-9 [24]. Item nine asks: “Over the last two weeks, how often have you been bothered by thoughts that you would be better off dead, or of hurting yourself in some way?” and is scored from 0 to 3. A score of 0 equals “not at all”, 1 equals “several days”, 2 equals “more than half the days”, and 3 equals “nearly every day”. A positive response to item 9 (i.e., ≥1) is a predictor of suicide attempt or death by suicide over the following year [14,15]. The item has previously been used to assess suicidal ideation in various immigrant populations [28,29]. 

### 2.3. Statistical Analysis

Suicidal ideation was analyzed partly as a dichotomous variable, not having or having suicidal ideation at any frequency. All values of “0” (*I have not at all been bothered by thoughts of being better off dead or of hurting myself during the last two weeks)* were coded as not having suicidal ideation (= 0). All values 1, 2, or 3, indicating being bothered by suicidal thoughts either several days, more than half the days, or nearly every day were coded as having suicidal ideation (= 1). Student t-test was used to compare means between those with and those without suicidal ideation. Pearson’s Chi-square and Spearman’s p were used to assess the strength of the association between suicidal ideation and distress severity level. A logistic regression, using the group with no distress (individuals scoring below 11) as the reference level, was performed to assess whether or not the odds of having suicidal ideation increased by distress severity. Some psychometrics were calculated in order to identify to what degree each severity level left individuals with suicidal ideation undetected. All analyses were performed with IBM SPSS Statistics for Windows (version 24, Armonk, NY: IBM Corp). 

## 3. Results

The mean value of the RHS-13 was 22.90 (*SD* 13.55), 95 % CI [21.72, 24.08], indicating a moderate level of distress for the group. Individuals without suicidal ideation also had a mean score in the moderate distress range: 18.60 (*SD* 12.52), 95% CI [17.26, 19.94], while individuals with suicidal ideation had a mean score in the severe range: 31.27 (*SD* 11.42), 95% CI [29.56, 32.99], *t*(508) = 11.15, *p* < 0.001, *d* = 1.04. One-third of the participants (n = 173) indicated that they had been bothered by suicidal ideation at some frequency. Eighteen percent of the total sample had these thoughts several days during the last two weeks, 8% had them more than half the days, and 8% had them nearly every day (see Table 1).

Participants experiencing no or minimal distress showed a low frequency of suicidal ideation. The frequency of suicidal ideation increased with increasing distress severity, χ^2^ = 94.17, *p* < 0.001, with Spearman’s ρ = 0.42, *p* < 0.001, showing a moderate association. 

The odds of having suicidal ideation were 19 times higher in those with severe distress than for those with no distress. However, the confidence interval was wide (ranging from 8.30 to 41.66, see Table 2). Because this large interval was likely indicative of too few cases of suicidal ideation in those with no distress (n = 7), we collapsed ”no distress” and ”mild distress” into one category for further analyses and compared it with the moderate and severe level. This operation resulted in an *OR* of 3.12, *p* < 0.001, 95% CI [1.67, 5.83] for the moderate level and in an *OR* of 8.09, *p* < 0.001, 95% CI [4.94, 13.25] for the severe level. The tightened confidence intervals suggest a more reliable result, although this obscures the lower but present risk in those with mild distress.

A cutoff of ≥11 for RHS-13 identified most people with any frequency of suicidal ideation (*N* = 166, test sensitivity = 0.96, negative predictive value; NPV = 0.94) leaving only 7 individuals (4%) undetected, of which 6 answered at the lowest frequency (several days) and one answered “more than half the days”. Likewise, the cutoff 18 renders a high sensitivity (0.86) and high NPV (0.87), failing to identify 25 out of the 173 positives, of which the majority had answered at the lowest frequency (*N* = 18), and only 2 at the highest frequency (nearly every day). At cutoff 25 sensitivity was 0.71 and the NPV was 0.82, failing to identify 51 individuals, however, capturing most that had suicidal ideation nearly every day (34 of 40). 

## 4. Discussion

This study showed a clear association between distress severity on the RHS-13 and suicidal ideation on one PHQ-9 item. Individuals with higher distress were more likely to report thoughts that they would be better off dead. Of the individuals with any suicidal ideation, 71% were found in the highest distress severity level. Furthermore, there was a moderate association between distress level and frequency of suicidal ideation, and the risk of an individual being suicidal increased significantly at each level of distress severity. The odds of an individual showing suicidal ideation was 19 times higher among individuals with severe distress than among individuals with no distress. This extends the findings by Bjärtå et al. [21], indicating that an RHS-13 score of ≥25 identifies individuals with severe distress who warrant acute intervention. Above the recommended cutoff score of 11, the RHS-13 identifies most individuals with any frequency of suicidal ideation with high sensitivity yet low specificity. Thus, a positive RHS-13 does not imply suicidal ideation, while a negative RHS-13 (<11) implies a low but present risk of suicidal ideation. A positive RHS-13 should, therefore, be followed by a suicide risk assessment. If resources are unlimited, this should be offered to all individuals scoring above 11. If resources are scarce, however, then only individuals scoring above 25 would be further assessed. 

The convenience sampling method limits generalizability of results to other populations. A second study limitation is that the sample consisted mainly of young men, which limits conclusions about females or other age groups. However, the sample is similar to the demographics of the population of asylum seekers in Sweden during this time period [30]. Another possible limitation concerns the choice of instruments. In the present study, suicidal ideation was assessed with item nine of the PHQ-9. In two consecutive studies including 84,418 patients [14] and 509,945 adult outpatients [15], this item was found to be a strong predictor of suicide attempt and death by suicide in the following two years. However, there are alternative reports indicating that the item is not accurate enough to be used as a suicide screener [31]. A meta-analysis assessing the general association between suicidal ideation and suicide questions the overall utility of suicidal ideation as a test for later suicide [32]. Although an association between expressed suicidal ideation and subsequent suicide was found, the positive predictive value of suicidal ideation for future death by suicide was low: 60% of the individuals who later died by suicide had not expressed any suicidal ideation. As corroboration, during development of the original version of the RHS, an item for suicidal ideation was included. However, complex analyses showed that this item did not predict distress as much as the current RHS-13 items. For this reason, and the fact that the utility of assessing suicidal ideation in all refugees has not been established, this item was not retained in the final RHS. The RHS should by no means be used to predict if an individual will end their life by suicide or not, but our current data suggest it might be useful to identify individuals at increased risk of suicide. In any case, frequent suicidal ideation is an evident marker of despair and it indicates that an individual is in need of mental health services.

The current study further validates the validity of the RHS-13 as a measure of distress and suggests that suicidal ideation is associated with distress severity but not in a highly reliable way: a clinician must not conclude that an individual with no or minimal distress or those *not* reporting any suicidal ideation is free from suicide risk. The previously reported elevated rates of death by suicide among refugees compel clinicians to pay attention to the risk of suicidal behaviors while assessing the mental health of refugees. In light of the present study, this seems to be crucial while encountering refugees with moderate to severe levels of distress but should not be omitted for individuals with lower levels of distress.

## 5. Conclusions

Asylum seekers with moderate or severe distress on the RHS-13 are more likely to endorse suicidal ideation than asylees with no or minimal distress. Although the relation between suicidal ideation and death by suicide is complex, these data call for an increased awareness of the suicide risk among resettled asylees and refugees.

## Figures and Tables

**Table 1 ijerph-16-02751-t001:** Prevalence of suicidal ideation by distress severity level on the RHS as assessed by item 9 in the PHQ-9. Number of respondents, percentage within each distress severity level and within each group is shown.

Frequency of Suicidal Ideation															
	Not at all			Several days			More than half the days			Nearly every day			Total Frequency		
	0			1			2			3			1 + 2 + 3		
RHS-13 level of distress	n	%row	%col	n	%row	%col	n	%row	%col	n	%row	%col	n	%row	%col
<11, n = 119	112	94	33	6	5	<1	1	<1	2	0	0	0	7	6	4
≥11; <18, n = 80	62	78	18	12	15	1	4	5	9	2	3	5	18	23	10
≥18; <25, n = 84	58	69	17	17	20	19	5	6	12	4	5	10	26	31	15
≥25, n = 227	105	46	31	55	24	61	33	15	77	34	15	85	122	54	71

Note: RHS-13 = Refugee Health Screener, 13-item version. PHQ-9 = Patient Health Questionnaire-9. %row = Percentage within each row/distress level. %col. = Percentage within each column/frequency of suicidal ideation.

**Table 2 ijerph-16-02751-t002:** Regression analysis of suicidal ideation at different levels of distress.

RHS-13 Level of Distress	β	*SE β*	Wald	*p*	*OR*	95 % CI
no/minimal <11			66.71	0.000		
mild ≥11	1.54	0.47	10.56	0.001	4.65	(1.84, 11.73)
moderate ≥18	1.97	0.46	18.71	0.000	7.17	(2.94, 17.51)
severe ≥25	2.92	0.41	50.39	0.000	18.59	(8.30, 41.66)
Constant	−2.77	0.39	50.65	0.000	0.06	

Note. RHS-13 = Refugee Health Screener 13. CI = confidence interval for odds ratio (*OR*).

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
