# Peer review of "Suicidal Ideation and Severity of Distress among Refugees Residing in Asylum Accommodations in Sweden"

_ijerph, 2019, doi:10.3390/ijerph16152751_

Round 1

Reviewer 1 Report

Line 31: ...suicide compared to the general referent population [10,11]. A recent Swedish report [12] shows high … (add "s" to the word show).

Line 39:  change the word "at" to "upon".

Line 53: Due to the unique characteristics and experiences of individuals from various cultures, it would be beneficial to know what groups (I.e. %) represented the population of refugees who participated in this study. ---If data is available.

Line 85: Suicidal ideation was measured using item nine of the PHQ-9 [24]. Use the term " using" rather than "with".

Line 152 and 153: The following sentence is confusing: "Thus, a positive test does not mean that individuals have suicidal ideations but a negative test means that they most likely don’t." Revise to increase clarity and flow of thought.

Line 153: Delete "s" from "ideations". It should be suicidal ideation.

Line 184: Replace the word “likelier” with “more likely”-increases flow of thought.

Line 185: Replace the word “show” with the term “exhibit”- word choice!

Reviewer 2 Report

This paper appears to present a secondary analysis of a previous published study regarding mental health of refugees in asylum centres in Sweden. The paper examined the association between mental health symptom severity and suicidal ideation, as measured with item 9 of the PHQ-9. The conclusion is justified, i.e., the risk of suicidal behaviour might be increased in those refugees with higher distress, which does not mean that there is no risk of suicidal behaviour in refugees with low distress. A few points need clarification.

Line 61: The researchers visited 13 locations with more than 30 refugees. Is that correct? Please rephrase: the N=13 is confusing in this sentence.

Line 61: What was the rationale to visit locations with at least 30 refugees? Why were other locations excluded? How could this selection have affected the results? Could this be a selection bias? If so, please add it to the limitations.

Line 62: I understand that participants filled out the survey on an iPad. The formulation “using the Qualtrics application” seems to complicate the sentence. Please rephrase.

Lines 69-70: Was there any difference in the scores of symptom severity or suicidal ideation between the group that had received a positive decision and the others? It could be conceivable that those with a positive decision might score better than the others? If you have not test this, how could it affect the interpretation of the study findings?

Lines 125 and 148: Please be consistent. It is either 18 or 19. The table said 18.59. Hence, 19 seems plausible.

Lines 169-170: The authors referred to RHS-15. However, earlier and throughout the manuscript the instrument was called RHS-13. The same lines also mentioned a suicidal ideation item that was not included. Please rephrase these sentences for clarity.

Overall, the manuscript needs proof reading and editing by a native English speaker. There are linguistic/grammatical errors, and switches between past and present tense of verbs.

Good luck with the revision.
